# ON THE GENERALIZATION OF SFT: A REINFORCEMENT LEARNING PERSPECTIVE WITH REWARD RECTIFICATION

**Yongliang Wu**[1][*]  **Yizhou Zhou**[2][*][†]  **Zhou Ziheng**[3]  **Yingzhe Peng**[1]  **Xinyu Ye**[4]
**Xinting Hu**[5]  **Wenbo Zhu**[6]  **Lu Qi**[7]  **Ming-Hsuan Yang**[8]  **Xu Yang**[1][‡]

[1]Southeast University    [2]Independent Researcher    [3]University of California, Los Angeles
[4]Shanghai Jiao Tong University    [5]Nanyang Technological University
[6]University of California, Berkeley    [7]Wuhan University    [8]University of California, Merced
`yongliang0223@gmail.com`, `zyz0205@hotmail.com`, `xuyang_palm@seu.edu.cn`

## ABSTRACT

In this work, we present a simple yet theoretically motivated improvement to Supervised Fine-Tuning (SFT) for the Large Language Model (LLM), addressing its limited generalization compared to reinforcement learning (RL). Through mathematical analysis, we reveal that standard SFT gradients implicitly encode a problematic reward structure that may severely restrict the generalization capabilities of model compared to RL. To rectify this, we propose Dynamic Fine-Tuning (DFT), stabilizing gradient updates for each token by dynamically rescaling the objective function with the probability of this token. With just a single-line change, the method outperforms standard SFT on multiple difficult benchmarks and base models, from math reasoning to code generation and multi-modal tasks, demonstrating improved generalization. Additionally, DFT achieves competitive results in offline RL settings, providing an effective yet streamlined alternative. By bridging theoretical insights with practical solutions, this work advances the state of SFT. The source code will be available at https://github.com/yongliang-wu/DFT.

## 1 INTRODUCTION

Supervised Fine-Tuning (SFT), which adapts models to expert demonstrations, has become the standard post-training paradigm for Large Language Models (LLMs) (Zhang et al., 2025c; 2024a; 2025b;a; Wang et al., 2025b). It enables efficient task adaptation and capability enhancement (Chung et al., 2024; Zhang et al., 2024c; Sanh et al., 2022; Ouyang et al., 2022; Chen et al., 2024; 2025a; Fang et al., 2025), and is popular for its ease of implementation and rapid acquisition of expert-like behaviors (Wei et al., 2022; Zhou et al., 2023). Despite these advantages, SFT often shows limited generalization compared to reinforcement learning (RL) (Chu et al., 2024; Ouyang et al., 2022; Christiano et al., 2017; Bai et al., 2022; Huan et al., 2025; Swamy et al., 2025). RL leverages explicit reward or verification signals to explore diverse strategies and thus generalizes better. However, RL requires substantial computation, careful hyperparameter tuning, and explicit reward signals—conditions often impractical in real-world settings (Schulman et al., 2017; Ouyang et al., 2022; Sheng et al., 2025; Strubell et al., 2019; Liu & Yin, 2024; Winsta, 2025). Moreover, RL can struggle to recover expert-like behaviors that SFT captures efficiently (Mandlekar et al., 2022; Chen et al., 2025c).

To exploit the complementary strengths of both approaches, many hybrid methods combine SFT with RL (Ouyang et al., 2022; Sheng et al., 2025; Rafailov et al., 2023; Liu et al., 2025; Qiu et al., 2025). Yet a key question remains: can SFT itself be fundamentally improved? This is crucial, as SFT remains the only viable option when datasets contain only positive demonstrations, with no negative samples or reward model available.

---

[*]Equal Contribution. [†]Project Leader. [‡]Corresponding Author.

In this work, we address this gap with a mathematical analysis of the connection between SFT and RL. We show that the gradient update in SFT can be interpreted as a form of policy gradient with a specific, implicitly defined reward under certain assumptions. Crucially, this reward is (i) sparse, and (ii) inversely proportional to the model's probability of expert actions (see equation 6). As a result, when the model assigns low probability to expert actions, the gradient becomes excessively large, yielding an ill-posed reward structure and unstable optimization (Pascanu et al., 2013; Yang et al., 2019).

Building on this insight, we propose Dynamic Fine-Tuning (DFT), a principled fix. Our method rescales the SFT objective at each token by its probability, canceling the distortion introduced by inverse-probability weighting. This reframing turns the SFT gradient from a potentially unstable and biased estimator into a more stable, more uniformly weighted update rule that behaves closer to an RL-style.

Empirically, DFT delivers substantial improvements. On the Qwen-2.5-Math series (Qwen Team et al., 2024b) fine-tuned with NuminaMath-CoT (LI et al., 2024), DFT yields gains several times larger than standard SFT. More importantly, unlike SFT, which often degrades on challenging benchmarks such as OlympiadBench (He et al., 2024), AIME 2024 (American Institute of Mathematics, 2024), and AMC 2023 (Mathematical Association of America, 2023), our method consistently improves performance and generalization. These improvements hold across models, scales, and data sizes (Table 1, Figure 1), and extend to code generation and multimodal reasoning (Tables 3, 4) (Zhao et al., 2025d; Luo et al., 2025; Li et al., 2025a;b) (Zhu et al., 2024b;c;a; 2020).

We further test DFT in off-policy RL settings (Table 2), where dense rewards are available (Levine et al., 2020). Our method not only outperforms offline RL approaches such as DPO (Rafailov et al., 2023) and RAFT (Dong et al., 2023; Ahn et al., 2024), but also achieves competitive or superior performance to online methods like GRPO and PPO on math tasks with Qwen2.5-Math-1.5B (He et al., 2025; Tan et al., 2026; Zhao et al., 2025b;a). Unlike these RL methods, DFT requires neither a reference model nor large batch sizes, making it a simpler and more resource-efficient alternative.

To understand its effect, we analyze token probability distributions after training (Figure 2). While traditional SFT uniformly pushes probabilities toward the training set, DFT selectively increases some while reducing others. In particular, the proportion of less strongly fitted tokens rises, suggesting improved regularization. We provide further discussion in Appendix A.3.

The contributions of this work are theoretical and practical. On the theoretical side, we mathematically establish LLM SFT as a special RL in policy gradient space, pinpoint the underlying reasons for the limited generalization of SFT, and derive a method to improve it. On the experimental side, we show that such a simple solution, just one line of code, can enhance the performance and generalization capabilities of SFT across various tasks and models.

## 2 RELATED WORK

The trade-off between supervised fine-tuning (SFT) and reinforcement learning (RL) is central to the alignment of large language models (Song et al., 2024; Chai et al., 2024; Song et al., 2025c;a; Xu et al., 2025; Song et al., 2025b; Zhu et al., 2025c;a). SFT is widely adopted due to its simplicity and efficiency in imitating expert demonstrations (Chung et al., 2024; Zhou et al., 2023; Wei et al., 2022), analogous to behavioral cloning in robotics (Sammut, 2011; Mandlekar et al., 2022). However, the literature consistently highlights its limitations, particularly the tendency to overfit and generalize poorly compared to RL, which leverages reward signals to discover more robust policies (Ouyang et al., 2022; Christiano et al., 2017; Bai et al., 2022; Swamy et al., 2025; Zhang et al., 2025d). A recent systematic comparison by Chu et al. (2024) across textual and visual domains confirms this distinction, concisely summarized as "SFT memorizes while RL generalizes." They further show that SFT remains indispensable as an initialization step, stabilizing output formatting prior to effective RL training. Nonetheless, RL faces significant practical hurdles, including computational expense, sensitivity to hyperparameters, and the requirement of an explicit reward function, all of which constrain its applicability (Schulman et al., 2017; Strubell et al., 2019; Sheng et al., 2025).

To combine the strengths of both paradigms, much recent work has pursued hybrid approaches. The most common strategy involves SFT pretraining followed by RL-based refinement with a learned reward model, as popularized by InstructGPT (Ouyang et al., 2022). More recent methods interleave

SFT and RL updates to improve stability and performance (Sheng et al., 2025; Liu et al., 2025; Qiu et al., 2025). Other approaches, such as Direct Preference Optimization (DPO) (Rafailov et al., 2023), bypass reward modeling entirely by directly optimizing policies on preference data, thereby unifying imitation and reinforcement signals within a single loss function. Chen et al. (2025b) introduce Negative-aware Fine-Tuning (NFT), which models incorrect generations via an implicit negative policy, enabling self-improvement without explicit feedback. While powerful, these methods rely on reward signals, preference pairs, or negative samples. They enrich the training pipeline but do not fundamentally improve SFT in its native setting, where only positive demonstrations are available. Our work instead focuses on enhancing SFT itself without requiring external feedback.

A complementary line of theoretical research seeks to unify SFT and RL under a common formalism. Du et al. (2025) reinterpret RLHF as a reward-weighted variant of SFT, preserving reliance on an explicit reward. Wang et al. (2025a) show that SFT can be cast as RL with an implicit reward, proposing adjustments such as smaller learning rates to manage the vanishing KL constraint. Abdolmaleki et al. (2025) analyze learning from both positive and negative feedback, studying how their balance affects convergence. Qin & Springenberg (2025) view SFT as a lower bound of RL and introduce importance weighting based on the data-generating policy. While these works establish connections between SFT and RL through weighting, they do not provide a precise mathematical equivalence between the SFT gradient and the offline policy gradient. Some methods approximate this connection in practice by reweighting training losses. For instance, MixCE (Zhang et al., 2023) combines the forward and reverse KL divergences to form a unified objective, while GOLD (Pang & He, 2021) adopts offline RL with demonstrations, introducing reliance on an unknown demonstration distribution $\pi_b$ and a restrictive $1/N$ assumption. Kantharaju & Sankar (2022) also provide a clear and insightful exposition of GOLD's motivation and mechanics from an alternative perspective, offering useful intuition for understanding its underlying design. Zhao et al. (2025c) provide a promising method to combine RL and SFT during training. In contrast, our work offers a more formal perspective on this connection, highlighting the role of the inverse-probability weighting term in shaping the difference between SFT and RL-like updates. This perspective motivates a simple adjustment: multiplying the loss by the model's token probability to neutralize the weighting.

Interestingly, our method modifies the standard cross-entropy (CE) loss in a way that inverts the weighting philosophy of the widely used Focal Loss (Lin et al., 2017). Specifically, our modified CE takes the form $-p\log(p)$, whereas focal loss is defined as $-(1-p)^\gamma \log(p)$. Focal Loss deliberately downweights well-classified samples to emphasize underrepresented or hard cases, whereas we deliberately downweight poorly classified samples to encourage generalization. This inversion reflects a fundamental shift in the LLM era: while underfitting was once a central challenge, overfitting and memorization now dominate, demanding a rethinking of objective design.

## 3 METHOD

### 3.1 PRELIMINARIES

**Supervised Fine-Tuning.** Let $\mathcal{D} = \{(x, y^\star)\}$ denote a corpus of expert demonstrations, where $y^\star$ is the complete reference response to the query $x$. SFT minimizes the sentence-level cross-entropy:

$$\mathcal{L}_{\mathrm{SFT}}(\theta) = \mathbb{E}_{(x,y^\star)\sim\mathcal{D}}\big[-\log\pi_\theta\big(y^\star \mid x\big)\big]. \tag{1}$$

Its gradient is:

$$\nabla_\theta\mathcal{L}_{\mathrm{SFT}}(\theta) = \mathbb{E}_{(x,y^\star)\sim\mathcal{D}}\big[-\nabla_\theta\log\pi_\theta\big(y^\star \mid x\big)\big]. \tag{2}$$

**Reinforcement Learning.** Let $y$ denote a response sampled from the policy $\pi_\theta(\cdot \mid x)$ for query $x$. Given a reward function $r(x, y) \in \mathbb{R}$, the policy objective is

$$J(\theta) = \mathbb{E}_{x\sim\mathcal{D}_x,\, y\sim\pi_\theta(\cdot|x)}\big[r(x,y)\big]. \tag{3}$$

Its policy gradient at the sentence level is

$$\nabla_\theta J(\theta) = \mathbb{E}_{x\sim\mathcal{D}_x,\, y\sim\pi_\theta(\cdot|x)}\big[\nabla_\theta\log\pi_\theta(y \mid x)\, r(x,y)\big]. \tag{4}$$

### 3.2 UNIFY SFT AND RL GRADIENT EXPRESSION

**Rewriting SFT Gradient as Policy Gradient via Importance Sampling.** The SFT gradient in equation 2 is taken under the *fixed* demonstration distribution. We convert it to an on-policy

expectation by inserting an importance weight that compares the expert (Dirac Delta) distribution with the model distribution.

$$\mathbb{E}_{(x,y^\star)\sim\mathcal{D}}\left[-\nabla_\theta\log\pi_\theta\big(y^\star\mid x\big)\right] = \mathbb{E}_{x\sim\mathcal{D}_x}\underbrace{\mathbb{E}_{y\sim\pi_\theta(\cdot\mid x)}\frac{\mathbf{1}[y=y^\star]}{\pi_\theta(y\mid x)}\left[-\nabla_\theta\log\pi_\theta\big(y\mid x\big)\right]}_{\text{resample + reweight}} \quad (5)$$

Define the auxiliary variables (importance sampling weight) as

$$w(y\mid x) = \frac{\mathbf{1}}{\pi_\theta(y\mid x)}, \quad r(x,y) = \mathbf{1}[y=y^\star].$$

Reorganizing equation 5 and rewriting it using the above auxiliary variables, we obtain the form

$$\nabla_\theta\mathcal{L}_{\text{SFT}}(\theta) = -\mathbb{E}_{x\sim\mathcal{D}_x,\,y\sim\pi_\theta(\cdot\mid x)}\left[w(y\mid x)\,\nabla_\theta\log\pi_\theta(y\mid x)\,r(x,y)\right]. \quad (6)$$

This form of the SFT gradient closely resembles the policy gradient in Equation equation 4. Under this formulation, conventional SFT can be interpreted as an on-policy gradient method, where the reward is a sparse indicator function matching the expert trajectory, but biased by an importance weighting term $1/\pi_\theta$. We emphasize that this RL-style characterization serves solely as a theoretical lens: both the analysis and subsequent modifications are developed within the RL framework, while the final method remains fully implementable in standard SFT form for computational efficiency. Detailed derivations are provided in Appendix A.2.

Due to the inherently sparse reward signal in the SFT setting, we identify the importance weight $1/\pi_\theta$ as a key contributor to SFT's generalization limitations compared to RL. When the model assigns low probability to the expert response, the resulting weight becomes excessively large, introducing an ill-posed reward landscape. This leads to disproportionately large gradients and training instability. The issue is compounded by the fact that the reward function $r(x,y) = \mathbf{1}[y=y^\star]$ is non-zero only for exact matches to the expert outputm causing optimization to overfit rare exact-match samples and weakening the model's ability to generalize beyond the training data.

### 3.3 PROPOSED METHOD

**Reward Rectification via Dynamic Reweighting.** To neutralize the skewed reward issue identified when viewing SFT under the RL objective, we dynamically reweight the reward by multiplying by a corrective inverse ratio given by the policy probability $1/w$. The resulting "dynamically fine-tuned" gradient is then

$$\nabla_\theta\mathcal{L}_{\text{SFT}}(\theta) = -\mathbb{E}_{x\sim\mathcal{D}_x,\,y\sim\pi_\theta(\cdot\mid x)}\left[\text{sg}(\frac{1}{w})\cdot w(y\mid x)\,\nabla_\theta\log\pi_\theta(y\mid x)\,r(x,y)\right]. \quad (7)$$

where $\text{sg}(\cdot)$ denotes the stop gradient operator, ensuring that gradients do not flow through the reward scaling term $w$. To facilitate transition to later equations, we directly write $1/w$ to be $\pi_\theta(y^\star\mid x)$ instead of $\pi_\theta(y\mid x)$ because the indicator function in equation 5 or equation 6 would leave all cases where $y\neq y^\star$ is 0. Now since the gradient does not flow, the corrected SFT loss also becomes a simple reweighted loss, called Dynamic Fine-tuning (DFT).

$$\mathcal{L}_{\text{DFT}}(\theta) = \mathbb{E}_{(x,y^\star)\sim\mathcal{D}}\left[-\text{sg}\left(\pi_\theta(y^\star\mid x)\right)\log\pi_\theta(y^\star\mid x)\right]. \quad (8)$$

However, in practice, computing importance weights over the entire trajectory can induce numerical instability. A common treatment of this issue is to simply apply importance sampling at the token level, as was adopted in PPO (Schulman et al., 2017). This leads to the final DFT loss version:

$$\mathcal{L}_{\text{DFT}}(\theta) = \mathbb{E}_{(x,y^\star)\sim\mathcal{D}}\left[-\sum_{t=1}^{|y^\star|}\text{sg}\left(\pi_\theta(y_t^\star\mid y_{<t}^\star,x)\right)\log\pi_\theta(y_t^\star\mid y_{<t}^\star,x)\right]. \quad (9)$$

Note that the reward of this corrected SFT (in RL form), i.e., DFT, now becomes 1 uniformly for all expert trajectory. This is akin to contemporary verification based reward approach

RLVR (DeepSeek-AI et al., 2025) that assigns uniform reward to all correct samples. Consequently, it avoids over-concentration on specific low-probability reference tokens, leading to more stable updates and improved generalization without introducing any additional sampling or reward models.

## 4 EXPERIMENTS

We design four groups of experiments to comprehensively evaluate DFT. We first study the standard SFT setting on mathematical reasoning tasks to establish its core advantage over SFT (Section 4.1). We then extend to an offline RL setting, comparing DFT with representative offline and online RL methods (Section 4.2). To test cross-domain robustness, we further examine DFT on code generation benchmarks (Section 4.3) and its applicability to multi-modal reasoning math datasets (Section 4.4).

### 4.1 MAIN EXPERIMENT - MATHEMATICAL REASONING TASK

To examine whether DFT can outperform vanilla SFT across tasks, architectures, and scales, we use mathematical reasoning as a representative testbed.

**Implementation details.**   To efficiently manage computational resources, We andomly sample 100,000 instances from the the NuminaMath-CoT dataset (LI et al., 2024) for training. We conduct experiments using multiple models, including Qwen2.5-Math-1.5B, Qwen2.5-Math-7B (Qwen Team et al., 2024a), LLaMA-3.2-3B, LLaMA-3.1-8B (Dubey et al., 2024), and DeepSeekMath-7B (Shao et al., 2024). Our implementation builds upon the verl framework (Sheng et al., 2025), using recommended SFT hyper-parameters. Specifically, we employ the AdamW optimizer with learning rates of $5 \times 10^{-5}$ for all models except the LLaMA-3.1-8B-Base, for which we adopt a lower learning rate of $2 \times 10^{-5}$. We set the mini-batch size to 256 and the maximum input length to 2048 tokens. The learning rate follows a cosine decay schedule with a warm-up ratio of 0.1. We evaluate on benchmarks including Math500 (Hendrycks et al., 2021), Minerva Math (Lewkowycz et al., 2022), Olympiad Bench (He et al., 2024), AIME 2024 (American Institute of Mathematics, 2024), and AMC 2023 (Mathematical Association of America, 2023) through the official Qwen2.5-Math evaluation pipeline (Qwen Team et al., 2024a). Each model uses the default chat template and Chain-of-Thought (CoT) prompting to stimulate step-by-step reasoning. All reported results represent average accuracy across 16 decoding runs, evaluated with a temperature of 1.0 and maximum generation length of 4096 tokens.

DFT consistently yields average performance improvements over base models compared to standard SFT across all benchmarks. Table 1 shows that, for Qwen2.5-Math-1.5B, DFT achieves an average gain of +15.66 points over the base model, which is over 5.9$\times$ larger than the +2.09 point improvement from SFT. This pattern generalizes across other model families and sizes: LLaMA-3.2-3B benefits from a +3.46 point gain with DFT, exceeding the SFT gain (+2.05) by approximately 1.4$\times$; LLaMA-3.1-8B achieves +10.02 from DFT, surpassing SFT's +5.33 by 1.88$\times$; DeepSeekMath-7B sees a +15.51 point improvement via DFT, which is 1.58$\times$ larger than SFT's +7.18; and Qwen2.5-Math-7B reaches a +15.90 point gain, nearly 3.8$\times$ higher than the SFT improvement of +2.37.

DFT demonstrates generalization and robustness, especially on challenging benchmarks where standard SFT yields minimal or even negative impact. For instance, on Olympiad Bench, SFT degrades performance for Qwen2.5-Math-1.5B, dropping accuracy from 15.88 to 12.63, while DFT boosts it to 27.08, +11.20 point improvement over base model. On AIME24, SFT reduces accuracy for Qwen2.5-Math-7B by 4.20 points (from 6.68 to 2.48), whereas DFT improves performance to 8.56, achieving a +1.88 point gain over the base model despite the difficulty of the benchmark. A similar trend is observed on AMC23. SFT reduces the performance of Qwen2.5-Math-1.5B from 19.38 to 18.75, while DFT raises it to 38.13, a +18.75 point gain over base. For Qwen2.5-Math-7B, SFT yields only a marginal improvement (+1.86), whereas DFT achieves a +17.04 point gain. These results underscore that DFT not only scales more effectively across models of varying capacities, but also exhibits better resilience on difficult reasoning tasks where traditional SFT struggles.

DFT exhibits better learning efficiency and faster convergence characteristics. Figure 1 reveals clear differences in learning dynamics between DFT and standard SFT on Qwen2.5-Math-1.5B across all math reasoning benchmarks. Compared to SFT, our method demonstrates three distinct advantages: (1) Faster convergence, achieving peak performance within the first 120 training steps

Table 1: Average@16 accuracy of five state-of-the-art large language models on mathematical reasoning benchmarks. The best performance of each model across benchmarks is bold.

| | Math500 | Minerva Math | Olympiad Bench | AIME24 | AMC23 | Avg. |
|---|---|---|---|---|---|---|
| LLaMA-3.2-3B | 1.63 | 1.36 | 1.01 | 0.41 | 1.56 | 1.19 |
| LLaMA-3.2-3B w/SFT | 8.65 | 2.38 | 2.06 | 0.00 | 3.13 | 3.24 |
| LLaMA-3.2-3B w/DFT | **12.79** | **2.84** | **2.90** | **0.83** | **3.91** | **4.65** |
| LLaMA-3.1-8B | 1.86 | 0.98 | 0.94 | 0.21 | 1.01 | 1.00 |
| LLaMA-3.1-8B w/SFT | 16.85 | 5.78 | 3.88 | 0.00 | 5.16 | 6.33 |
| LLaMA-3.1-8B w/DFT | **27.44** | **8.26** | **6.94** | **0.41** | **12.03** | **11.02** |
| DeepSeekMath-7B | 6.15 | 2.15 | 1.74 | 0.21 | 2.97 | 2.64 |
| DeepSeekMath-7B w/SFT | 26.83 | 7.26 | 6.33 | 0.41 | 8.28 | 9.82 |
| DeepSeekMath-7B w/DFT | **41.46** | **16.79** | **15.00** | **1.24** | **16.25** | **18.15** |
| Qwen2.5-Math-1.5B | 31.66 | 8.51 | 15.88 | 4.16 | 19.38 | 15.92 |
| Qwen2.5-Math-1.5B w/SFT | 43.76 | 13.04 | 12.63 | 1.87 | 18.75 | 18.01 |
| Qwen2.5-Math-1.5B w/DFT | **64.89** | **20.94** | **27.08** | **6.87** | **38.13** | **31.58** |
| Qwen2.5-Math-7B | 40.12 | 14.39 | 17.12 | 6.68 | 27.96 | 21.25 |
| Qwen2.5-Math-7B w/SFT | 53.96 | 16.66 | 18.93 | 2.48 | 26.09 | 23.62 |
| Qwen2.5-Math-7B w/DFT | **68.20** | **30.16** | **33.83** | **8.56** | **45.00** | **37.15** |

on most benchmarks; (2) Better early-stage performance, with DFT already outperforming best final accuracy of SFT within the first 10–20 steps; and (3) Higher sample efficiency, consistently requiring fewer updates to reach relatively optimal results. This accelerated convergence shows that the dynamic reweighting mechanism in DFT leads to more informative gradient updates, guiding the model toward high-quality solutions early in training. It also suggests that DFT helps avoid the optimization plateaus or noise-prone regions often encountered in standard SFT, thereby enabling more efficient acquisition of complex mathematical reasoning patterns.

We also report the results of parameter-efficient fine-tuning (PEFT) training setting (Hu et al., 2022) and training on the OpenR1-Math dataset (Hugging Face, 2025) with better quality in Appendix A.6 and Appendix A.5, respectively. Comparison and Discussion with the concurrent method iw-SFT (Qin & Springenberg, 2025) is provided in Appendix A.4.

## 4.2 EXPLORATORY EXPERIMENT - OFFLINE RL SETTING

Equation 7 shows that SFT suffers from reward sparsity, since in a constructed dataset each query $x$ has only a single reference answer $y^\star$. From the perspective of RL, RFT/RAFT (Dong et al., 2023; Ahn et al., 2024) can be viewed as alleviating the sparse reward issue by effectively increasing reward density, thereby enhancing model performance. Motivated by this observation, we conduct an exploratory study applying DFT in an offline RL setting, where the reward sparsity problem is inherently less severe compared to standard SFT, to further validate the effectiveness.

**Implementation details.** We sample responses for 100,000 math questions using a temperature of 1.0 and generate four responses per question from the base model itself. Correct responses are identified using math verify and retained as training data, resulting in approximately 140,000 examples. For DPO training, we construct 100,000 positive–negative preference pairs from the generated responses. We compare DFT with representative offline RL methods, including DPO (Rafailov et al., 2023) and RFT (Dong et al., 2023; Ahn et al., 2024), as well as online RL methods PPO (Schulman et al., 2017) and GRPO (Shao et al., 2024). For RFT and DFT, the training setup follows the configuration in Section 4.1. For DPO, we use the ms-swift (Zhao et al., 2024) with a learning rate of $1 \times 10^{-6}$, batch size of 128, and a warmup ratio of 0.05. For PPO and GRPO, training is performed using the verl (Sheng et al., 2025) with a learning rate of $1 \times 10^{-6}$, batch size of 256, and a warmup ratio of 0.1. We set the number of response $n = 4$ for GRPO.

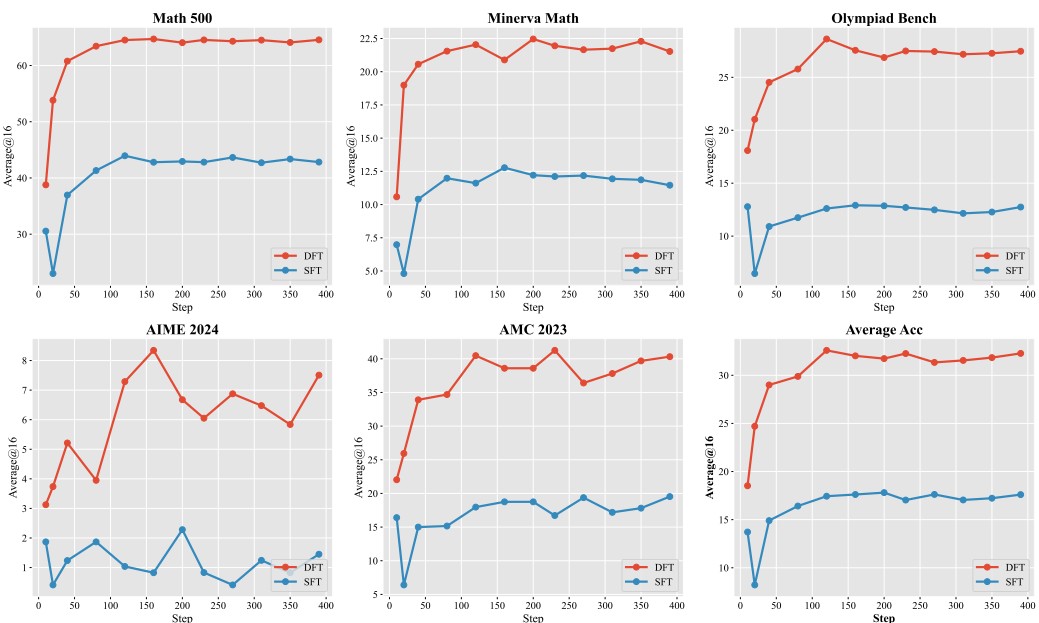

Figure 1: Accuracy progression for Qwen2.5-Math-1.5B across mathematical benchmarks, illustrating faster convergence and better performance achieved by DFT relative to SFT.

Table 2: Evaluation results on mathematical reasoning benchmarks in an offline reinforcement learning setting using reward signals from rejection sampling. The best performance is in bold.

|  | Setting | Math500 | Minerva Math | Olympiad Bench | AIME24 | AMC23 | Avg. |
|---|---|---|---|---|---|---|---|
| Qwen2.5-Math-1.5B w/DFT | SFT | 64.89 | 20.94 | 27.08 | 6.87 | 38.13 | 31.58 |
| Qwen2.5-Math-1.5B w/DPO | Offline | 46.89 | 11.53 | 22.86 | 4.58 | 30.16 | 23.20 |
| Qwen2.5-Math-1.5B w/RFT | Offline | 48.23 | 14.19 | 22.29 | 4.37 | 30.78 | 23.97 |
| Qwen2.5-Math-1.5B w/PPO | Online | 56.10 | 15.41 | 26.33 | 7.50 | 37.97 | 28.66 |
| Qwen2.5-Math-1.5B w/GRPO | Online | 62.86 | 18.93 | 28.62 | **8.34** | 41.25 | 32.00 |
| Qwen2.5-Math-1.5B w/DFT | Offline | **64.71** | **25.16** | **30.93** | 7.93 | **48.44** | **35.43** |

DFT demonstrates competitive performance in the offline RL setting, outperforming both offline and online RL baselines. Table 2 shows DFT achieves an average score of 35.43, exceeding the best offline method RFT by +11.46 points, and even outperforming the strongest online RL algorithm GRPO by +3.43 points. Specially, on Math500, DFT scores 64.71, slightly ahead of GRPO (62.86) and better than PPO (56.10) and RFT (48.23). The gains are also notable on more challenging benchmarks: on AMC23, DFT achieves 48.44, a +7.19 point margin over GRPO and a +17.66 point gain over RFT. Similarly, on Minerva Math, DFT reaches 25.16, outperforming GRPO by +6.23 points, PPO by +9.75, and all offline baseline methods.

These results highlight the strength of DFT as a simple yet effective fine-tuning strategy. Despite its lack of iterative reward modeling or environment interaction, it provides a stronger learning signal than both offline methods like DPO/RFT and online policy optimization algorithms like PPO/GRPO in certain scale train set. This suggests that DFT can serve as a more efficient and scalable alternative to traditional RL pipelines, particularly in domains where preference supervision is available but reward modeling or online response sampling is expensive or impractical.

### 4.3 EXPLORATORY EXPERIMENT - CODE GENERATION TASK

**Implementation details.** We adopt UltraFeedback (Cui et al., 2024) as the training dataset. From this corpus, we sample 10,000 prompts and, for each prompt, select the response with the highest average score to perform supervised fine-tuning (SFT) (Du et al., 2025). Model performance is assessed on three widely used code generation benchmarks: HumanEval (Chen et al., 2021), Hu-

Table 3: Performance of various models on code generation benchmarks. The best performance for each benchmark is highlighted in bold.

| | HumanEval | | MultiPL-E | | | | | | | | |
| | HE | HE+ | Python | C++ | Java | PHP | TS | C# | Bash | JS | Avg. |
|---|---|---|---|---|---|---|---|---|---|---|---|
| Qwen2.5-3B | 43.3 | 36.0 | 43.29 | 40.99 | 37.34 | 37.89 | **47.17** | **43.04** | 24.68 | 45.96 | 40.05 |
| Qwen2.5-3B w/SFT | 41.5 | 34.8 | 42.07 | 42.24 | 37.97 | 37.27 | 43.40 | 41.77 | 20.25 | **47.83** | 39.10 |
| Qwen2.5-3B w/DFT | **45.7** | **39.0** | **45.73** | **44.72** | **41.77** | **45.34** | 42.14 | **43.04** | **27.85** | 44.10 | **41.84** |
| Qwen2.5-Coder-3B | 52.4 | 42.7 | 51.83 | 53.42 | 46.20 | 47.20 | 54.09 | 55.06 | 25.32 | **54.04** | 48.39 |
| Qwen2.5-Coder-3B w/SFT | 51.8 | 43.9 | 51.22 | 51.55 | 48.10 | 54.66 | **59.12** | 51.27 | **34.18** | **54.04** | 50.52 |
| Qwen2.5-Coder-3B w/DFT | **56.7** | **50.0** | **57.32** | **54.66** | **51.27** | **58.39** | 58.49 | **60.76** | 31.01 | 53.42 | **53.16** |
| Qwen2.5-Coder-7B | 62.2 | 53.0 | 63.41 | 63.98 | 53.16 | 59.01 | 62.89 | 59.49 | 39.24 | 60.87 | 57.76 |
| Qwen2.5-Coder-7B w/SFT | 54.9 | 48.8 | 54.88 | 64.60 | 51.27 | **62.11** | 68.55 | 60.76 | 33.54 | **65.22** | 57.62 |
| Qwen2.5-Coder-7B w/DFT | **67.7** | **59.8** | **67.68** | **67.70** | **54.43** | 60.87 | **70.44** | **65.19** | **48.73** | 63.35 | **62.30** |

Table 4: Performance comparison across different multi-modal reasoning benchmarks. The best performance on each benchmark is highlighted in bold.

| | MathVerse | | | | MathVision | WeMath |
| | Vision Only | Vision Intensive | Vision Dominant | Overall | | |
|---|---|---|---|---|---|---|
| Qwen2.5-VL-3B | 28.81 | 30.96 | 31.60 | 33.83 | 21.25 | 4.10 |
| Qwen2.5-VL-3B w/SFT | 30.96 | 33.63 | 32.74 | 35.66 | 21.02 | 23.33 |
| Qwen2.5-VL-3B w/DFT | **32.49** | **35.91** | **33.50** | **37.54** | **22.30** | **23.71** |

manEval+ (Liu et al., 2023), and MultiPL-E (Cassano et al., 2023). Training is conducted for one epoch with a learning rate of $5 \times 10^{-5}$, a warm-up ratio of 0.05, and a batch size of 16.

Table 3 shows DFT achieves improvements in most cases compared to both base models and SFT. For Qwen2.5-3B, DFT raises HumanEval from 43.3 to 45.7 and HumanEval+ from 36.0 to 39.0, with the MultiPL-E average also increasing from 40.05 (base) and 39.10 (SFT) to 41.84. Similar trends are observed for Qwen2.5-Coder-3B, where DFT improves HumanEval to 56.7 and HumanEval+ to 50.0, outperforming both base and SFT. For Qwen2.5-Coder-7B, DFT reaches 67.7 on HumanEval, 59.8 on HumanEval+, and 62.3 average on MultiPL-E, surpassing SFT by +12.8, +11.0, and +4.7 points respectively. The overall trend demonstrates that DFT generally provides stronger performance across different models and languages.

## 4.4 EXPLORATORY EXPERIMENT - MULTI-MODAL REASONING

**Implementation details.** We use the WeThink dataset (Yang et al., 2025) for training. The model is fine-tuned using LLaMA-Factory (Zheng et al., 2024) and evaluated with VLMEvalKit (Duan et al., 2024). We train the model for 1 epoch with a learning rate of 5e-5. To comprehensively assess reasoning capabilities, we adopt a suite of multi-modal reasoning benchmarks including MathVerse (Zhang et al., 2024b), MathVision (Wang et al., 2024), and WeMath (Qiao et al., 2024) for evaluation.

DFT achieves consistent improvements over base models and SFT across all multi-modal reasoning benchmarks. Table 4 shows, on MathVerse, DFT boosts Qwen2.5-VL-3B from 33.83 to 37.54 average accuracy, outperforming the SFT gain of only +1.83 by +3.71 points. Consistent improvements are observed across all major vision-related subcategories. On MathVision, DFT improves performance from 21.25 (base) to 22.30, exceeding SFT which fails to provide gains (21.02). On WeMath, SFT already yields a +19.23 point gain, but DFT pushes performance slightly further to 23.71, maintaining superiority over both base and SFT. These results indicate that DFT not only strengthens text-only reasoning but also extends effectively to multi-modal domains.

Table 5: Comparison of weighting strategies on mathematical reasoning benchmarks.

| | Math500 | Minerva Math | Olympiad Bench | AIME24 | AMC23 | Avg. |
|---|---|---|---|---|---|---|
| Qwen2.5-Math-1.5B | 31.66 | 8.51 | 15.88 | 4.16 | 19.38 | 15.92 |
| Sentence-Level Weighting | 31.26 | 8.05 | 16.47 | 3.12 | 19.84 | 15.75 |
| Geometric-Mean Weighting | 42.87 | 12.34 | 13.03 | 1.23 | 16.56 | 17.21 |
| Token-Level Weighting | **64.89** | **20.94** | **27.08** | **6.87** | **38.13** | **31.58** |

### 4.5 Limitations of DFT: A Case Study on Factual Knowledge

While DFT consistently outperforms SFT on reasoning-heavy tasks, it may not always be the better choice, particularly in factual knowledge domains. We conduct an exploratory experiment on the Natural Questions dataset (Kwiatkowski et al., 2019), which consists of real-user, open-domain factual queries grounded in Wikipedia articles.

In this setting, we find that SFT improves performance from 31.24% to 36.62%, while DFT unexpectedly reduces it to 30.14%. This result reveals an important limitation of DFT: because it reweights samples based on the model's own confidence, it tends to reinforce the model's existing beliefs. When the model lacks sufficient factual knowledge, such reinforcement may hinder effective learning instead of facilitating it.

This case suggests that DFT is most effective when the task aligns well with the model's prior competence, such as logical reasoning or structured prediction. In contrast, when the objective is to absorb new factual information, especially in domains beyond the model's current capabilities, SFT remains a more reliable and stable fine-tuning strategy.

### 4.6 An Empirical Comparison with Sentence-Level Weighting

Our framework applies confidence-based weighting at the token level. While this design was primarily motivated by numerical stability, we also compared it against two sentence-level variants to better understand their behavior.

The first variant uses the full sequence probability to scale the loss. However, these values are extremely small in practice, making the loss nearly uninformative and producing a highly skewed weight distribution that is difficult to tune. To address this, we also evaluated a geometric-mean variant inspired by GSPO (Zheng et al., 2025), which rescales sentence probabilities to avoid numerical collapse. Although this version is more stable, it still provides a weak training signal and offers limited performance gains.

As shown in Table 5, both sentence-level strategies lead to minimal changes over the base model, while our token-level formulation delivers substantial and consistent improvements, raising average accuracy from 15.92 to 31.58. These results demonstrate that token-level weighting provides a more reliable optimization signal and significantly stronger empirical performance.

### 4.7 Analysis of Probabilities

To understand how the model trained by DFT is different from SFT and other RL methods, we look into the token probability distribution of the model's output over the training set in Figure 2. SFT tends to uniformly increase token probabilities, shifting the entire distribution towards higher confidence, but mainly targeting the lower and lowest probability tokens. The highest probability token portion barely increases. In stark contrast, DFT exhibits a polarizing effect: it significantly boosts the probabilities of a subset of tokens while actively suppressing the probabilities of others. This leads to a bimodal distribution, with more tokens occupying both the highest and lowest probability bins. Other RL methods such as DPO, GPPO and PPO show the same trend as DFT, although the scale is much milder than it. We look into the words that belong to the lowest probability bin, and find that they are generally the conjunctive words or punctuations such as 'the', 'let', ',', '.' etc. These results suggest that for robust learning, models should not attempt to fit all tokens with uniform confidence. It may be beneficial to deprioritize fitting tokens that serve grammatical

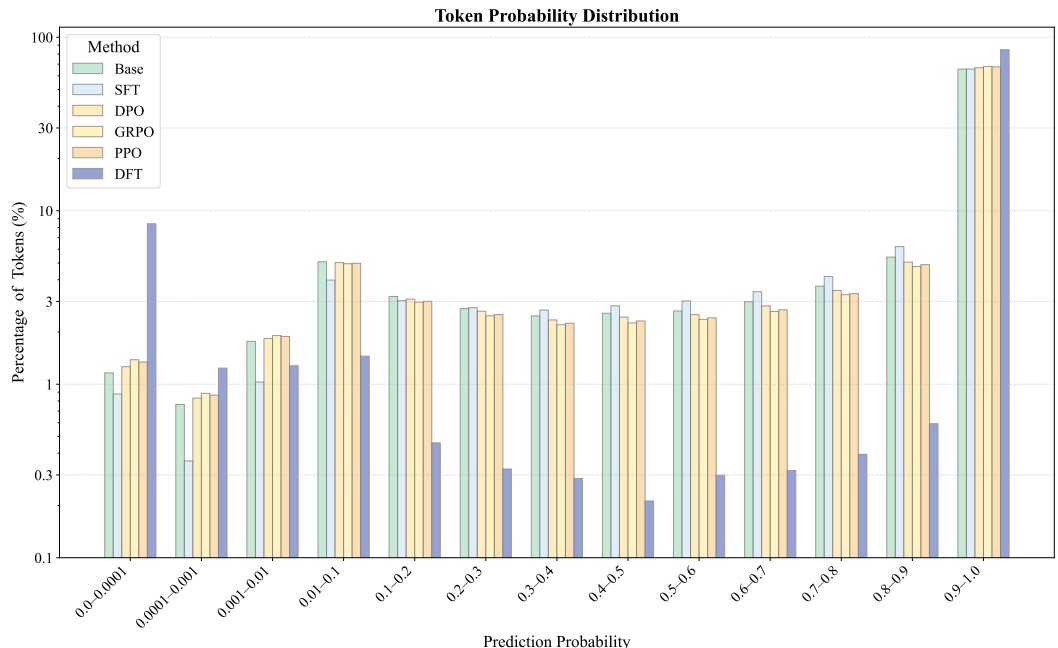

Figure 2: Token probability distributions on the training set before training and after fine-tuning with DFT, SFT, and various RL methods. A logarithmic scale is used on the y-axis for clarity.

functions rather than carrying primary semantic content. This concept is analogous to human pedagogy, where students are taught to focus on substantive concepts rather than perfecting the usage of common connective words. Further analysis can be found in Appendix A.3.

## 5  CONCLUSION

In this work, we revisit the well-known generalization gap between SFT and RL. We offer a theoretical perspective showing that the standard SFT gradient can be interpreted as a policy gradient with an ill-posed, implicitly defined reward inversely related to model confidence. This formulation helps explain the instability and limited generalization observed in SFT training. Motivated by this analysis, we introduce DFT, a simple yet effective method that dynamically reweights the SFT loss using the token probability. This one-line change improves gradient stability and leads to better generalization. Our empirical results show that DFT consistently improves over standard SFT across a range of models and challenging mathematical reasoning tasks. Beyond supervised settings, we adapt DFT to offline RL scenarios and find that it outperforms several established online and offline RL baselines, suggesting broader applicability. Overall, this work contributes both a refined understanding of SFT's limitations and a lightweight, practical method that helps bridge the gap to more complex RL-based approaches.

**Limitations.**  While our experiments demonstrate the effectiveness of DFT on mathematical reasoning benchmarks and code generation tasks, the evaluation scope remains limited. We have not yet assessed its performance on broader task categories or with larger-scale LLM, which we leave for future exploration. Moreover, DFT can not offer universal benefits across all scenarios. In domains that primarily involve the acquisition of factual knowledge, conventional SFT still remains the most efficient approach. DFT may also not be an ideal choice for hard examples or domains underrepresented in the training data, since it assigns low initial probabilities to such samples, reducing their learning weight. Our aim is not to assert that DFT universally outperforms SFT, but rather to offer a new perspective on objective design by analyzing the distinction between RL and SFT. Besides, an important future direction is to explore non-uniform or quality-aware reward assignments for demonstrations.

ACKNOWLEDGEMENTS

Supported by Jiangsu Province Carbon Peak Carbon Neutrality Science and Technology Innovation Special Fund Project (Grant No. BT2025029), National Natural Science Foundation of China (Grant No. 62576091), and Big Data Computing Center of Southeast University.

ETHICS STATEMENT

This work adheres to the ICLR Code of Ethics. Our study does not involve human subjects, personally identifiable information, or proprietary data. All datasets used, including NuminaMath, OpenR1-Math, UltraFeedback, and WeThink, are publicly available and documented in the appendix. The proposed method is a simple training strategy that modifies gradient computation for improved generalization. It does not introduce any new capabilities that could cause harm, nor does it enable misuse beyond the standard capabilities of existing large language models. We are not aware of any potential risks related to bias, fairness, or security that arise specifically from the method proposed. Nonetheless, we acknowledge that like any fine-tuning strategy, DFT may inherit biases present in the underlying data or model, and future research may explore safeguards for these scenarios. No conflicts of interest, legal compliance issues, or sponsorship-related influences are present in this work.

REPRODUCIBILITY STATEMENT

We have taken multiple steps to ensure the reproducibility of our work. All datasets used in our experiments are publicly available and properly cited in the main text and appendix. Training configurations, including model architectures, hyperparameters, optimizers, and evaluation settings, are described in detail in Section 4 and Appendi A.5-A.6. Theoretical claims, including the equivalence between SFT and policy gradient, are formally derived in Appendix A.2. Experimental results include multiple model scales, tasks, and training settings to validate robustness. A complete implementation of our method is included in the supplementary material, along with scripts for reproducing all reported results. We will release the full source code and training logs upon publication to further support reproducibility.

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

# A APPENDIX

## A.1 USAGE OF LLM

We employ LLM primarily as writing assistants to refine and polish the manuscript. Their usage was limited to improving clarity, coherence, and presentation, while all conceptual and experimental contributions remain original.

## A.2 DETAILED DERIVATION OF EQUATION (5)

We start from the SFT gradient in Equation (2):

$$\nabla_\theta \mathcal{L}_{\text{SFT}}(\theta) = \mathbb{E}_{(x,y^\star)\sim\mathcal{D}}\big[ - \nabla_\theta \log \pi_\theta(y^\star \mid x)\big]. \tag{1}$$

For each query $x$, the expectation over expert demonstrations $(x, y^\star)$ can be written explicitly as a summation over all possible outputs $y$:

$$\mathbb{E}_{(x,y^\star)\sim\mathcal{D}}\big[ - \nabla_\theta \log \pi_\theta(y^\star \mid x)\big] = \mathbb{E}_{x\sim\mathcal{D}_x} \sum_y \mathbf{1}[y = y^\star]\big[ - \nabla_\theta \log \pi_\theta(y \mid x)\big]. \tag{2}$$

We insert the model distribution $\pi_\theta(y \mid x)$, which allows us to express the summation in terms of importance weights:

$$\mathbb{E}_{x\sim\mathcal{D}_x} \sum_y \pi_\theta(y \mid x) \cdot \frac{\mathbf{1}[y = y^\star]}{\pi_\theta(y \mid x)} \big[ - \nabla_\theta \log \pi_\theta(y \mid x)\big]. \tag{3}$$

Here, the term $\frac{\mathbf{1}[y=y^\star]}{\pi_\theta(y|x)}$ serves as an importance weight comparing the expert (Dirac delta) distribution with the model's distribution.

The summation over $y$ can now be rewritten as an expectation under the policy distribution $y \sim \pi_\theta(\cdot \mid x)$:

$$\mathbb{E}_{x\sim\mathcal{D}_x} \mathbb{E}_{y\sim\pi_\theta(\cdot|x)} \left[ \frac{\mathbf{1}[y = y^\star]}{\pi_\theta(y \mid x)} \big( - \nabla_\theta \log \pi_\theta(y \mid x)\big)\right]. \tag{4}$$

Thus, we obtain Equation (5):

$$\mathbb{E}_{(x,y^\star)\sim\mathcal{D}}\big[ - \nabla_\theta \log \pi_\theta(y^\star \mid x)\big] = \mathbb{E}_{x\sim\mathcal{D}_x} \mathbb{E}_{y\sim\pi_\theta(\cdot|x)} \left[ \frac{\mathbf{1}[y = y^\star]}{\pi_\theta(y \mid x)} \big( - \nabla_\theta \log \pi_\theta(y \mid x)\big)\right]. \tag{5}$$

This derivation shows that the SFT gradient can be expressed as an on-policy policy gradient with importance sampling, where the expert demonstration distribution is reweighted relative to the model distribution.

## A.3 DISCUSSIONS AND INSIGHTS

**Gradient Analysis of DFT.** We now analyze the gradient induced by the DFT surrogate loss. Recall the sequence-level definition:

$$\mathcal{L}_{\text{DFT}}(\theta) = - \text{sg}\big(\pi_\theta(y^\star \mid x)\big) \, \log \pi_\theta(y^\star \mid x), \tag{10}$$

where $\text{sg}(\cdot)$ denotes the stop-gradient operator. Since the stop-gradient blocks backpropagation, the detached probability $\text{sg}(\pi_\theta(y^\star \mid x))$ is treated as a constant during differentiation. Consequently, the gradient becomes

$$\nabla_\theta \mathcal{L}_{\text{DFT}} = - \text{sg}\big(\pi_\theta(y^\star \mid x)\big) \frac{1}{\pi_\theta(y^\star \mid x)} \nabla_\theta \pi_\theta(y^\star \mid x) \tag{11}$$

$$= - \left( \frac{\text{sg}(\pi_\theta(y^\star|x))}{\pi_\theta(y^\star|x)} \right) \nabla_\theta \pi_\theta(y^\star \mid x). \tag{12}$$

Since $\mathrm{sg}(\pi_\theta(y^\star \mid x))$ equals $\pi_\theta(y^\star \mid x)$ in the forward pass, the prefactor is numerically equal to $1$. Therefore,

$$\nabla_\theta \mathcal{L}_{\mathrm{DFT}} = -\nabla_\theta \pi_\theta(y^\star \mid x). \tag{13}$$

This shows that DFT is mathematically equivalent to directly maximizing the model probability of the target token, rather than its log-probability as in cross-entropy.

For cross-entropy, the loss is

$$\mathcal{L}_{\mathrm{CE}}(\theta) = -\log \pi_\theta(y^\star \mid x),$$

yielding gradient

$$\nabla_\theta \mathcal{L}_{\mathrm{CE}} = -\frac{1}{\pi_\theta(y^\star \mid x)} \nabla_\theta \pi_\theta(y^\star \mid x).$$

Thus both CE and DFT share the same gradient direction but differ in scaling: CE amplifies updates for low-probability targets (factor $1/\pi$), while DFT applies a uniform factor $1$. As a result, DFT avoids the instability caused by excessively large gradients on unlikely expert tokens, providing more conservative and stable updates.

From the reinforcement learning perspective, the reward of DFT becomes uniformly $1$ across all expert trajectories, equivalent to a verification-style objective that treats all correct references equally. From the optimization perspective, DFT trades off aggressive fitting of rare tokens for better stability and calibration. Practically, this explains why DFT often yields smoother training and stronger generalization, while maintaining alignment with the pre-training distribution.

**Learning from Noisy Data.** DFT offers a simple yet effective approach, prompting us to reflect on why it might actually work. One intuitive explanation lies in its ability to learn from noisy data (Freund, 2009). Sasaki & Yamashina (2020) propose an imitation learning algorithm for learning from noisy demonstrations, based on the core idea of avoiding the fitting of data that is difficult to model, as such data is likely to originate from suboptimal behaviors, i.e., noise. Their method introduces a weighted behavioral cloning objective, where the weights are derived from a previously trained policy's confidence in each action. Similarly, the weighting mechanism in DFT shares the same intuition, but instead of relying on a fixed old policy model to compute confidence scores, it uses a single policy model to perform confidence-based weighting on-the-fly during training.

## A.4 Comparision with Concurrent work iw-SFT

We include a concurrent method, Importance-Weighted SFT (iw-SFT) (Qin & Springenberg, 2025), for comparison. All training settings follow those reported in the original paper, except that we set the number of training epochs to 1.

As shown in Table 6, DFT achieves higher average accuracy than iw-SFT on most model families: LLaMA-3.2-3B (+2.39), LLaMA-3.1-8B (+4.15), DeepSeekMath-7B (+3.34), and Qwen2.5-Math-1.5B (+1.30). Although iw-SFT outperforms our method on Qwen2.5-Math-7B (+2.45), this improvement is not consistent across datasets. In particular, for LLaMA-3.2-3B, iw-SFT underperforms standard SFT on Math500 (5.13 vs. 8.65) and AMC23 (2.03 vs. 3.13). Similarly, for LLaMA-3.1-8B, iw-SFT results in worse performance than SFT on Minerva Math (4.31 vs. 5.78) and AMC23 (7.34 vs. 8.28). In contrast, DFT consistently improves upon both the base model and SFT across nearly all datasets, including those where iw-SFT fails. These results underline better generalization ability of DFT in diverse mathematical reasoning scenarios. Moreover, iw-SFT incurs additional computational overhead by requiring a separate reference model to compute importance weights, whereas DFT dynamically derives its own weighting directly from the token probabilities of model, resulting in a more efficient training procedure.

We also compare against iw-SFT under the offline setting, as shown in Table 7. While iw-SFT performs competitively on certain datasets, achieving 60.80 on Math500 and 44.21 on AMC23, its overall average performance (31.86) remains below that of our method by +3.57 points. Moreover, iw-SFT shows only modest improvements compared to its standard SFT counterpart, with an average score of 31.86 in the offline RL setting versus 30.28 with SFT (+1.58). In contrast, DFT achieves a larger gain of +4.76 (from 30.67 to 35.43). These results indicate that iw-SFT provides limited benefits from reward supervision under offline constraints, whereas DFT is able to more effectively incorporate such signals, leading to better generalization and higher task performance.

Table 6: Comparison with concurrent work iw-SFT on math benchmarks. DFT outperforms the iw-SFT in most settings across model families and benchmarks. The best performance is bold.

| | Math500 | Minerva Math | Olympiad Bench | AIME24 | AMC23 | Avg. |
|---|---|---|---|---|---|---|
| LLaMA-3.2-3B w/iw-SFT | 5.13 | 2.63 | 1.51 | 0.00 | 2.03 | 2.26 |
| LLaMA-3.2-3B w/DFT | **12.79** | **2.84** | **2.90** | **0.83** | **3.91** | **4.65** |
| LLaMA-3.1-8B w/iw-SFT | 18.21 | 4.31 | 4.31 | 0.20 | 7.34 | 6.87 |
| LLaMA-3.1-8B w/DFT | **27.44** | **8.26** | **6.94** | **0.41** | **12.03** | **11.02** |
| DeepSeekMath-7B w/iw-SFT | 35.32 | 8.75 | 11.11 | 0.61 | **18.28** | 14.81 |
| DeepSeekMath-7B w/DFT | **41.46** | **16.79** | **15.00** | **1.24** | 16.25 | **18.15** |
| Qwen2.5-Math-1.5B w/iw-SFT | 59.38 | 17.08 | 26.82 | **8.13** | **40.00** | 30.28 |
| Qwen2.5-Math-1.5B w/DFT | **64.89** | **20.94** | **27.08** | 6.87 | 38.13 | **31.58** |
| Qwen2.5-Math-7B w/iw-SFT | **70.28** | 25.70 | **34.46** | **16.46** | **51.09** | **39.60** |
| Qwen2.5-Math-7B w/DFT | 68.20 | **30.16** | 33.83 | 8.56 | 45.00 | 37.15 |

Table 7: Evaluation results on five mathematical reasoning benchmarks in the offline reinforcement learning setting using rejection-sampling–based reward signals, compared against iw-SFT.

| | Setting | Math500 | Minerva Math | Olympiad Bench | AIME24 | AMC23 | Avg. |
|---|---|---|---|---|---|---|---|
| Qwen2.5-Math-1.5B w/iw-SFT | SFT | 59.38 | 17.08 | 26.82 | 8.13 | 40.00 | 30.28 |
| Qwen2.5-Math-1.5B w/DFT | SFT | 62.50 | 22.94 | 26.87 | 7.31 | 33.75 | 30.67 |
| Qwen2.5-Math-1.5B w/iw-SFT | Offline | 60.80 | 18.13 | 27.83 | **8.33** | 44.21 | 31.86 |
| Qwen2.5-Math-1.5B w/DFT | Offline | **64.71** | **25.16** | **30.93** | 7.93 | **48.44** | **35.43** |

## A.5 EXPLORATORY EXPERIMENT - OPENR1-MATH TRAINING DATASET

Inspired by DeepSeek-R1 DeepSeek-AI et al. (2025), several studies have attempted to train open-source models to reproduce its reasoning capabilities (Hugging Face, 2025). To this end, a high-quality dataset, OpenR1-Math-220k (Hugging Face, 2025), was constructed, where the prompts are drawn from NuminaMath 1.5 and the off-policy reasoning traces are generated by DeepSeek-R1. LUFFY (Yan et al., 2025) further filtered out sequences longer than 8192 tokens as well as those verified incorrect by Math-Verify, resulting in about 45k prompts paired with off-policy reasoning traces. We adopt this dataset as the training corpus for SFT. All training details remain the same as previous experiments, except that the number of epochs is set to 3.

As shown in Table 8, training on OpenR1-Math-220k consistently improves performance, and the use of higher-quality annotations yields additional gains. SFT on this dataset increases the average accuracy of Qwen2.5-Math-1.5B by +13.24 points compared to the base model, while DFT provides a further +9.03 gain, resulting in a total improvement of +22.27. These results suggest that DFT remains effective even when applied on top of high-quality training data, highlighting its potential as a general fine-tuning paradigm.

## A.6 EXPLORATORY EXPERIMENT - PEFT TRAINING SETTING

To investigate whether DFT remains effective under parameter-efficient fine-tuning (PEFT) settings with limited compute, we apply DFT using LoRA adapters across two model families: LLaMA-3.2-3B and Qwen2.5-Math-1.5B. All training configurations remain identical to previous full-parameter experiments, except that LoRA is enabled with rank=8 and alpha=16.

As shown in Table 9, DFT provides consistent improvements over both base and SFT baselines under LoRA-based PEFT. For Qwen2.5-Math-1.5B, DFT increases the average accuracy from 15.92 (base) and 16.87 (SFT) to 32.90. For LLaMA-3.2-3B, DFT achieves a gain of +3.44 over SFT (from 1.19 to

Table 8: Performance training on the OpenR1-Math-220k dataset. The best score for each benchmark is in bold.

|  | Math500 | Minerva Math | Olympiad Bench | AIME24 | AMC23 | Avg. |
|---|---|---|---|---|---|---|
| Qwen2.5-Math-1.5B | 31.66 | 8.51 | 15.88 | 4.16 | 19.38 | 15.92 |
| Qwen2.5-Math-1.5B w/SFT | 61.60 | 20.29 | 24.27 | 4.16 | 35.47 | 29.16 |
| Qwen2.5-Math-1.5B w/DFT | **71.76** | **27.00** | **33.48** | **9.79** | **48.91** | **38.19** |

Table 9: Performance on five mathematical reasoning benchmarks using LoRA for training. The best score of each model across benchmarks is highlighted in bold.

|  | Math500 | Minerva Math | Olympiad Bench | AIME24 | AMC23 | Avg. |
|---|---|---|---|---|---|---|
| LLaMA-3.2-3B | 1.63 | 1.36 | 1.01 | **0.41** | 1.56 | 1.19 |
| LLaMA-3.2-3B w/SFT | 4.88 | 1.56 | 1.68 | 0.00 | 2.66 | 2.56 |
| LLaMA-3.2-3B w/DFT | **11.13** | **5.18** | **3.87** | 0.00 | **2.97** | **4.63** |
| Qwen2.5-Math-1.5B | 31.66 | 8.51 | 15.88 | 4.16 | 19.38 | 15.92 |
| Qwen2.5-Math-1.5B w/SFT | 41.47 | 10.85 | 11.56 | 1.45 | 17.03 | 16.87 |
| Qwen2.5-Math-1.5B w/DFT | **64.85** | **22.58** | **28.45** | **5.84** | **40.78** | **32.90** |

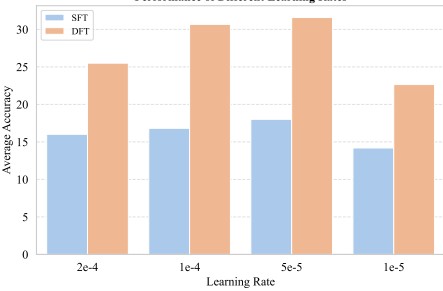 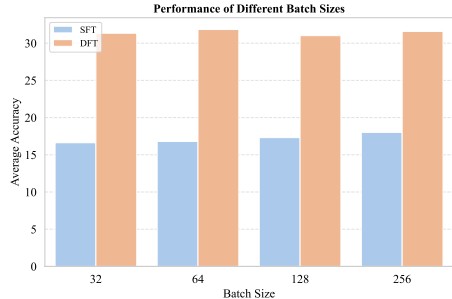

Figure 3: Ablation study of training hyper-parameters, learning rates and batch size, for DFT and SFT on Qwen2.5-Math-1.5B model.

4.63). These results indicate that DFT can serve as an effective fine-tuning strategy in low-resource or compute-constrained settings, where full model updates are not practical.

## A.7 TRAINING HYPER-PARAMETERS ABLATION

To assess the robustness and sensitivity of our approach (DFT) with respect to key training hyper-parameters, we conduct an ablation study focused on learning rate and batch size, using the Qwen2.5-Math-1.5B base model. This analysis aims to answer two central questions: (1) Is the performance gap between DFT and SFT due to a suboptimal hyperparameter configuration in SFT? (2) How sensitive are both methods to changes in learning rate and batch size?

We evaluate both DFT and SFT across four learning rates: 2e-4, 1e-4, 5e-5, and 1e-5. As shown in Figure 3 (left), both methods exhibit a certain degree of sensitivity to the learning rate. DFT consistently outperforms SFT under all configurations, suggesting that the performance gap cannot be attributed solely to suboptimal hyperparameter choices in SFT. For both methods, intermediate learning rates (1e-4 and 5e-5) yield the best results, while both lower (1e-5) and higher (2e-4) values lead to noticeable degradation.

We further assess the impact of batch size, sweeping values from 32 to 256. As shown in Figure 3 (right), both DFT and SFT exhibit relatively stable performance across the full range of batch sizes.

While minor fluctuations are observed, there is no consistent trend indicating that larger or smaller batches significantly affect final accuracy. This suggests that batch size is not a dominant factor for either method in this setup, and that default values may suffice in practice.

