# OpenReview forum: "On the Generalization of SFT: A Reinforcement Learning Perspective with Reward Rectification"
_ICLR.cc/2026/Conference — ICLR 2026 Poster_

### Official Review · Reviewer_8xgv · 2025-10-26

**Soundness:** 3
**Presentation:** 2
**Contribution:** 3
**Rating:** 6
**Confidence:** 4

**Summary:**

This paper presents Dynamic Fine-Tuning (DFT), a method that improves the generalization of Supervised Fine-Tuning (SFT). The authors first provide a analysis by framing SFT from a reinforcement learning perspective, identifying that its implicit reward is inversely proportional to the model's confidence, which leads to unstable training. DFT rectifies this by reweighting the SFT loss with the token probability, a simple modification that is shown through extensive experiments to deliver performance gains across diverse models and tasks.

**Strengths:**

- The theoretical motivation is clear, connecting the SFT gradient to policy gradients and pinpointing the problematic inverse-probability weighting as the root cause of poor generalization.
- The proposed DFT is simple to implement, requiring only a minor modification to the standard SFT loss function, yet it yields substantial empirical improvements.
- The evaluation is comprehensive, spanning multiple reasoning domains including mathematics, code generation, and multi-modal tasks.

**Weaknesses:**

- My main concerns lie in the potential conceptual limitations and unintended consequences of the core reweighting mechanism. The strategy of down-weighting low-probability tokens encourages the model to ignore what it finds hard. While this appears to prevent overfitting on noisy or rare patterns in the presented experiments, the paper does not adequately explore the boundary conditions under which this approach might fail. A more thorough discussion or ablation study on when this *hard example ignorance* becomes detrimental would strengthen the paper's claims.
- The authors mention avoiding numerical instability as a reason for token-level weighting but lacks detail on how it handles token probabilities that are extremely close to zero, which could cause vanishing gradients for those specific tokens.
- The implicit assumption that a uniform reward of 1 is optimal for all expert demonstrations might be an oversimplification, as it treats demonstrations of varying quality, complexity, or importance equally.

**Questions:**

- How does the DFT method influence the calibration of the model's output probabilities?
- The decision to apply weighting at the token-level was justified by stability. Did the authors investigate applying a sequence-level probability as the weight, perhaps with safeguards like clipping, and how did its performance compare?

---

> ### Author Response · Authors · 2025-11-20
> **Response to Reviewer 8xgv (1/2)**
>
> Dear Reviewer 8xgv,
>
> We appreciate the your positive assessment of our derivations, methodological contributions, and experimental results. Your recognition provides valuable encouragement and helps guide the further refinement of our work. Here is our response:
>
> **W1. Potential Limitations of DFT**
>
> We appreciate the reviewer for raising this important conceptual concern. In practice, it is often unclear whether a low-probability example corresponds to meaningful difficulty or merely reflects imperfections or inconsistencies in the data. Since DFT does not explicitly distinguish between these cases, we acknowledge that its reweighting mechanism carries the risk of under-training samples to which the model assigns very low initial probability.
>
> This limitation becomes especially pronounced in scenarios where learning requires the acquisition of new factual knowledge, which refers to information that lies entirely outside the base model’s pre-training distribution. Such cases can naturally be viewed as a form of hard samples, because the model has no prior exposure to the underlying facts and therefore assigns uniformly low probability to the correct answers. Under these conditions, DFT’s down-weighting of low-probability tokens suppresses the gradients that are necessary for the model to absorb this new knowledge within a limited number of training steps.
>
> To more concretely examine this behavior, we conducted a small exploratory study on the Natural Questions dataset [1], which evaluates open-domain factual recall. In this setting, SFT improves accuracy from 31.24% to 36.62%, whereas DFT reduces it to 30.14%. We have added these results to the updated version of the paper. This experiment provides direct empirical support for the reviewer’s intuition: when the supervision signal requires the model to absorb previously unseen factual content, which represents truly hard samples, the gradient underweighting introduced by DFT can indeed have detrimental effects.
>
> One possible mitigation is to introduce a clipped lower bound on the DFT weighting itself (e.g., enforcing a minimum weight such as 0.1), ensuring that difficult examples are never entirely down-weighted and that their signal remains influential during training.
> Finally, because our work reformulates SFT from an RL perspective, DFT naturally inherits RL-like behaviors, most notably, a dependence on the underlying model’s competence. When the base model is weak in a particular domain, the reweighting mechanism may amplify this weakness, similar to how RL methods struggle when rewards lie outside the model’s prior knowledge.
>
> We thank the reviewer again for highlighting this important point and have revised the paper to more explicitly discuss these boundary conditions and limitations.
>
> **W2. Does a near-zero token probability lead to vanishing gradients?**
>
> We acknowledge that extremely small token probabilities can indeed lead to vanishing gradients under token-level weighting. This is an inherent weakness of the approach. However, this issue can be mitigated by applying the clipping strategy discussed above, introducing a lower bound on the weighting (e.g., ensuring it does not fall below 0.1) so that even low-probability tokens retain sufficient gradient signal.
>
> **W3. Is it problematic that all expert demonstrations receive a uniform reward of 1?**
>
> We would like to clarify that this uniform reward of 1 is not an assumption introduced by our method, but rather an inherent assumption shared by standard SFT and by most existing RL formulations used in language model training. Under the SFT setup, all demonstrations are treated equally regardless of their quality, difficulty, or importance. Similarly, in mathematical reasoning tasks, RL methods typically assign a reward of 1 for a correct answer, again implicitly assuming uniform demonstration value.
>
> We fully acknowledge that, in principle, demonstrations may deserve different reward magnitudes, and exploring non-uniform or quality-aware reward assignments is an important future direction. In fact, DFT is also compatible with any specific non-uniform reward design which indicates different reward assignment priors. However, this lies beyond the scope of the present work. We will explicitly clarify this point in the revision and discuss it as a promising avenue for future research.

---

> ### Author Response · Authors · 2025-11-20
> **Response to Reviewer 8xgv (2/2)**
>
> **Q1. How does DFT affect calibration of model probability?**
>
> Figure 2 in our paper illustrates how DFT reshapes the model’s probability distribution. SFT tends to uniformly raise token probabilities, shifting the entire distribution toward higher confidence, mainly by increasing the probabilities of low- and mid-confidence tokens. The highest-probability region changes only marginally. In contrast, DFT induces a polarization effect: it substantially amplifies the probabilities of some tokens while explicitly suppressing others. This produces a bimodal pattern, with more tokens moving into both the highest and lowest probability bins.
>
> Examining the tokens pushed into the lowest-confidence bin, we find that they are predominantly connective or grammatical items such as “the,” “let,” commas, and periods. This suggests that robust learning does not require uniformly increasing confidence across all tokens. Instead, deprioritizing purely functional tokens, those that contribute little semantic content, may be beneficial for improving model behavior and calibration.
>
> **Q2. Token-level vs. sequence-level weighting**
>
> Thank you for the insightful suggestion. For numerical stability, we adopt token-level scaling, a strategy also analyzed and used in prior work [2,3]. We examined sequence-level weighting by multiplying the loss with the full sequence probability, but found that these values are extremely small in practice: the maximum across the training set is around 4e–7, while the median drops to approximately 4e–57. Such tiny magnitudes drive the loss toward zero and introduce substantial variance, making optimization highly unstable and clipping thresholds difficult to tune [3].
>
> To further investigate, we also evaluated a geometric-mean variant inspired by GSPO [4], which avoids the extreme numerical collapse of raw sequence probabilities. While more stable, this variant still provides a very weak learning signal and yields negligible improvements.
>
> We have incorporated the full comparison in the revised manuscript (Table 8). As shown below, both sentence-level variants perform similarly to the baseline, whereas token-level weighting delivers substantial and consistent gains.
>
> | Method                     | Math500 | Minerva Math | Olympiad Bench | AIME24 | AMC23 | Avg.  |
> |----------------------------|---------|--------------|-----------------|--------|-------|-------|
> | Qwen2.5-Math-1.5B          | 31.66   | 8.51         | 15.88           | 4.16   | 19.38 | 15.92 |
> | Sentence-Level Weighting   | 31.26   | 8.05         | 16.47           | 3.12   | 19.84 | 15.75 |
> | Geometric-Mean Weighting   | 42.87   | 12.34        | 13.03           | 1.23   | 16.56 | 17.21 |
> | **Token-Level Weighting**  | **64.89** | **20.94**    | **27.08**       | **6.87** | **38.13** | **31.58** |
>
> We hope our clarifications help resolve the concerns you raised.
>
> **Reference**
>
> [1] Natural questions: a benchmark for question answering research
>
> [2] Offline Reinforcement Learning: Tutorial, Review, and Perspectives on Open Problems
>
> [3] Text Generation by Learning from Demonstrations
>
> [4] Group Sequence Policy Optimization

---

> > ### Comment · Reviewer_8xgv · 2025-11-27
> >
> > I recognize the efforts made by the authors, and the rebuttal has addressed my concerns.

---

> > > ### Author Response · Authors · 2025-11-27
> > >
> > > Thank you for your thoughtful comments, and we’re glad to hear that our response has addressed your concerns.

---

### Official Review · Reviewer_d8uu · 2025-10-28

**Soundness:** 3
**Presentation:** 2
**Contribution:** 2
**Rating:** 6
**Confidence:** 4

**Summary:**

Proposes a modification to supervised finetuning (SFT) for language models, called Dynamic Fine-Tuning (DFT). Instead of minimizing the standard cross-entropy loss, DFT weights each gradient of an output by the probability of that output. This change is motivated by framing SFT as a reinforcement learning method. Empirical evaluations demonstrate that DFT outperforms SFT across several benchmarks and base models.

**Strengths:**

1. Relatively well-written and easy to follow.

2. The idea behind DFT is intuitive and conceptually simple.

3. Experiments demonstrate that DFT leads to substantial performance gains over SFT across several models and settings, covering mathematical reasoning, code generation, and multi-modal tasks.

**Weaknesses:**

1. The theoretical motivation behind DFT is imprecise and includes several unsubstantiated claims.

    - The dependence of SFT on $1 / \pi_\theta (y | x)$ is fake, in the sense that it is obtained by multiplying and dividing by $\pi_\theta (y | x)$. In particular, it is not true that the gradient of SFT grows unboundedly due to the term $1 / \pi_\theta (y | x)$ in Equation (6), as this term cancels out with the expectation.

    - The claim of SFT having sparse rewards does not make much sense to me. In SFT, you are guaranteed access to the output $y*$ with the optimal reward of 1 (according to the formulation in the paper). Such a claim about sparseness would be true if one would actually sample outputs from $\pi_\theta$ until landing on $y^*$, but this is not what SFT does.

    - In several places (e.g., line 51 and lines 167 to 171), a connection is being made between large gradient norm, unstable training, and overfitting. Regardless of the fact that, as mentioned above, the gradient norm of SFT does not grow unboundedly when $\pi_{\theta} (y^{*}|x)$ is small, why would large gradient norms lead to overfitting? When making such claims you need to either ground them in the literature or theoretical evidence. Otherwise, it is best to refrain from making them or clearly hedge them as intuition to not mislead readers.

    To be clear, if a method works well in practice, I do not believe that theoretical motivation is strictly necessary. There are plenty of heuristics that are useful in practice, yet lack theoretical backing. However, if one does choose to motivate their method theoretically, that motivation should be sound and clearly separate intuition from formal arguments, which can be backed by results in the literature or new results. Another downside of imprecise theoretical motivation is that it may hide the true reason for why DFT improves upon SFT.


2. The paper lacks a discussion of potential limitations of DFT. For example, weighting outputs by their current probability should induce a tendency to prefer high probability outputs, thereby sharpening the model’s distribution. This suggests that the performance of DFT may highly depend on the initial model, to a greater extent than SFT. Does DFT indeed improve over SFT across the board, or when finetuning on domains further away than the model's current capabilities SFT works better? Are there any other potential limitations of DFT or do the authors always recommend using DFT over SFT?



Review Summary and Recommendation
---

Overall, DFT seems to bring substantial benefits compared to SFT across several practically relevant settings. The fact that such an improvement can be achieved by a simple (in a good way) tweak to SFT is quite remarkable. I therefore tend toward recommending acceptance. However, I do strongly suggest either toning down some of the imprecise and unsubstantiated claims made regarding SFT (see weaknesses portion of the review) or grounding them in a formal analysis or empirical evaluation.


Additional (More Minor) Comments
---

1. In Equation (7) there is a minor issue with the definition of DFT: $L_{SFT}$ was defined as an expectation over the training data, while here it is treated as if it is the loss over a single example $(x, y^*)$.

2. Typo in Equation (8): I believe there should be a minus sign there (i.e., the objective is to minimize the negative log probability and not the log probability).

**Questions:**

--

---

> ### Author Response · Authors · 2025-11-20
> **Response to Reviewer d8uu (1/2)**
>
> Dear reviewer d8uu,
>
> Thank you for your positive assessment of our derivations, methodology, and experimental results. Here is our response:
>
> **W1: Theoretical Motivation**
>
> A central goal of our work is to reinterpret SFT through an on-policy RL lens. Our claims about instability, sparsity, and implicit reward structure are not assertions about SFT’s literal numerical behavior, but observations that arise only after translating the SFT objective into a RL formulation. This perspective shift, while conceptually simple, is what allows us to identify the structural mismatches between SFT and RL optimization that DFT is designed to address. We will revise the paper to clarify that these points are conceptual insights from the RL interpretation, rather than statements about the operational mechanics of SFT itself.
>
> **(1) On the role of the 1/πθ(y|x) term in the SFT gradient**
>
> We appreciate the your clarification and fully agree that standard SFT does not suffer from numerical instability caused by an explicit 1/πθ(y∣x) term. Our derivation of SFT in a RL expectation is not intended to imply the existence of such instability, nor to claim that SFT gradients grow unbounded in practice. Rather, this reformulation serves purely as an analytical device for exposing the implicit reward structure induced by maximum-likelihood training.
>
> To put it more concretely, we employ this reformulation to reveal that SFT is exactly an on policy rl but with obvious flaws. Fixing these flaws leads to a more robust RL process. All the analysis and fixes are done within RL. But obviously we can cast it back to SFT during the implementation phase for efficiency (actually one can choose to implement it in a pure rl way). Reverting to the SFT formulation during the analysis stage would obscure these insights.
>
>  In the revision, we will make this distinction explicit and highlight that the RL-form expression is used solely as a conceptual interpretation tool, without suggesting any actual numerical dependence on 1/πθ(y∣x) in SFT.
>
> **(2) On the claim that SFT exhibits sparse rewards**
>
> We agree that SFT supervision is not sparse in the operational sense, models directly observe y*, and no sampling from πθ is required. Our statement concerns the implicit reward distribution revealed by the RL interpretation of the SFT gradient, where reward mass is concentrated exclusively on the demonstrated target tokens and effectively zero elsewhere. Under this interpretation, the supervision signal behaves like a highly peaked reward, which contrasts with the richer feedback structures commonly used in RL.
>
> Moreover, in practical large-scale training, SFT data is rarely perfectly curated, and a single demonstration cannot always be regarded as the unique optimal output. This motivates techniques such as rejection sampling [3], which generate multiple candidate outputs and select high-quality ones, effectively densifying the reward signal when viewed from an RL perspective.
>
> **(3) On the relation between gradient magnitude, instability, and overfitting**
>
> Thank you for pointing this out. We consulted prior literature that discusses the relationship between gradient magnitude and training instability [1–2]. At the same time, we acknowledge that large gradients do not necessarily lead to overfitting. We have removed those statements in the revision.

---

> ### Author Response · Authors · 2025-11-20
> **Response to Reviewer d8uu (2/2)**
>
> **W2: Insufficient discussion of potential limitations.**
>
> Thank you for the suggestion. We have included a section on Limitations in the submission version.
>
> "While our experiments demonstrate the effectiveness of DFT on mathematical reasoning benchmarks and code generation tasks, the evaluation scope remains limited. We have not yet assessed its performance on broader task categories or with larger-scale LLM, which we leave for future exploration. Moreover, DFT can not offer universal benefits across all scenarios. In domains that primarily involve the acquisition of factual knowledge, conventional SFT still remains the most efficient approach. Our aim is not to assert that DFT universally outperforms SFT, but rather to offer a new perspective on objective design by analyzing the distinction between RL and SFT."
>
> Your analysis is absolutely correct. Since the motivation behind DFT is to reinterpret SFT from an RL perspective, it naturally inherits some RL-like behavior, most notably, its dependence on the underlying base model’s performance. Because DFT reweights samples according to the model’s own confidence, it cannot be expected to outperform SFT in all settings. As we mention in the discussion, in tasks with stable and well-defined “golden” targets, such as factual knowledge acquisition, standard SFT remains the more effective approach.
>
> To verify this point, we also conduct a small exploratory experiment on the Natural Questions dataset [4], which evaluates open-domain factual recall. In this setting, SFT improves accuracy from 31.24% to 36.62%, whereas DFT decreases it to 30.14%. We have updated the results in the paper. Thank you for your suggestion.
>
>
> **Minor Weakness**
>
> We have already addressed this issue in the revised version of the paper. Thank you for pointing it out!
>
> We hope our clarifications help resolve the concerns you raised!
>
> **Reference**
>
> [1] On the difficulty of training Recurrent Neural Networks
>
> [2] A Mean Field Theory of Batch Normalization
>
> [3] RAFT: Reward rAnked FineTuning for Generative Foundation Model Alignment
>
> [4] Natural questions: a benchmark for question answering research

---

> > ### Comment · Reviewer_d8uu · 2025-11-24
> >
> > Thank you for the response. I have read it and the other reviews carefully. In particular, I appreciate the addition of the limitations paragraph, which helps clarify where DFT is expected to help and where it may not.
> >
> > I will keep my initial (weakly) positive assessment of the paper due to this addition and the empirical benefits that DFT seems to bring compared to SFT across several practically relevant settings. Though, it is worth noting that the most major weakness discussed in the review still stands: the theoretical motivation behind DFT is imprecise and includes several unsubstantiated claims. These claims are mostly echoed and not resolved in the authors' response, as detailed below. Specifically, it is currently unclear whether the theoretical motivation captures why DFT is helpful, to the extent that it potentially obfuscates the reason that DFT improves upon SFT. I therefore strongly recommend hedging and adding required nuance to the current motivation provided for DFT.
> >
> > > Our claims about instability, sparsity, and implicit reward structure are not assertions about SFT’s literal numerical behavior, but observations that arise only after translating the SFT objective into a RL formulation.
> >
> > If the claims on instability, sparsity of rewards, and implicit reward structure do not hold for the way SFT actually operates, then any arguments on shortcomings of SFT based on them are invalid.
> >
> > > …we employ this reformulation to reveal that SFT is exactly an on policy rl but with obvious flaws.
> >
> > I do not agree with this statement. One can divide and multiply any objective by $\pi_\theta (y | x)$. Does that mean it is equivalent to an on-policy RL objective? Not in any meaningful way because that $\pi_\theta (y |x)$ cancels out. I believe saying that this reformulation reveals that SFT is “exactly an on-policy RL” objective is imprecise.

---

> > > ### Author Response · Authors · 2025-11-27
> > >
> > > Thank you for your detailed feedback and for maintaining a positive assessment of our work.
> > >
> > > It is well known that RL can be more suitable than SFT in certain scenarios, and often leads to stronger generalization when applicable. However, in many practical large-scale settings only expert demonstrations are available, making standard RL infeasible. Our goal is therefore not to criticize SFT per se, but to explore whether one can recover some of the benefits of RL even in the demonstration-only regime. The reformulation was introduced purely as a mathematical device to highlight the differences between the two objectives and to motivate why reweighting could make SFT behave more like RL under this constraint.
> > >
> > > Therefore, if our writing gives an impression that SFT is having problem, it is simply under the context of trying to bring SFT closer to RL in where RL performs better. We are not intending to make a general claim that such gaps are intrinsic problem of SFT.  I believe this might be the root cause of our discussion so far.
> > >
> > > Thank you again for raising your concern. It helps us to realize how our writing could lead to a wrong impression and how it should be improved. We’ve made adjustments to clarify our intent and hope the current version better reflects our position.
> > >
> > > Thank you again for your comments!

---

### Official Review · Reviewer_JU9o · 2025-10-30

**Soundness:** 2
**Presentation:** 3
**Contribution:** 2
**Rating:** 6
**Confidence:** 3

**Summary:**

The paper observes that supervised fine-tuning (SFT) encodes a sparse reward that limits its generalization ability when compared to reinforcement learning (RL). Due to the sparse nature of the reward, using SFT may lead to optimization difficulty and/or poor generalization. The paper introduces a method called dynamic fine-tuning (DFT) that reweighs the SFT gradient to avoid the problem observed with SFT. The effectiveness of DFT over SFT is analyzed via experiments on several language models. Additional results are provided with other setups including coding and multimodal models.

**Strengths:**

- The analytical arguments are clear and presented in an easy-to-follow manner. The proposed fix via weighing the gradient follows naturally from the analytical argument.

- Experimental results that show improvement over SFT models are fairly extensive. These experiments cover a variety of language model families (Llama, Qwen, DeepSeek).

- The impelemtnation included in the appendix is easy to understand and use by a general ML practitioner.

**Weaknesses:**

-While the analysis is good, there may be other papers that have made observations about sparse rewards and done the derivation perhaps under a different guise. I appreciate the authors referring to GOLD (Pang & He, 2021) in the paper as I learned about this method as well. A follow up ICLR Blog Post does a derivation that looks similar to what is included in the paper (https://iclr-blog-track.github.io/2022/03/25/text-gen-via-lfd). So the derivation, while insightful, is perhaps not as valuable as the empirical results.

- Table 1 reports SFT vs DFT results. I wonder why there are no results available with RL (SFT + RL, DFT + RL perhaps?). I am curious to understand whether DFT-based models can be effective when used to further train with RL. So the claim on DFT surpassing RL may need to be qualified as I do not see any RL methods in Table 1

- Similar comment can be made for other setups like coding and multimodal models considered in the paper

**Questions:**

- Please check the ICLR Blog Post I shared above and see if that merits being cited in the paper. While the analysis is still useful my suggestion is to focus on empirical results as the main contribution in the paper

- I would like to understand why there are no RL results in Table 1 (and others).

- Are DFT-based models RL-able?

Overall, this is a well written and simple to follow paper. I look forward to interacting with the authors, other reviewers and AC to get a deeper understanding of this work and make a decision.

The following paper may be of interest to the authors:
"Simplify RLHF as Reward-Weighted SFT: A Variational Method" https://arxiv.org/abs/2502.11026

(Note: I am sharing the above in the spirit of helping out a colleague in the research community. Citing the above is not necessary for me to recommend an acceptance.)

---

> ### Author Response · Authors · 2025-11-20
> **Response to Reviewer JU9o**
>
> Dear Reviewer JU8o,
>
> Thank you for your recognition of the derivation, methodology, and experimental results presented in our paper. Here is our response:
>
> **Q1. On the Novelty of the Derivation and Response to the ICLR Blog Post**
>
> We appreciate your reference to the blog post, which offers a thorough discussion of the motivation, methodology, and experiments of the GOLD paper, as well as insightful perspectives on future research directions.
>
> Despite having a similar gradient form, the underlying motivation behind our derivation is fundamentally different:
>
> - GOLD believes that since, in text generation, the learning objective and the evaluation metric differ, one should use OFFLINE RL for expert demonstrations; but this naturally brings in an unknown demonstration distribution and requires a strong assumption with behavior policy = 1/N.
>
> - DFT’s idea is that, if we explain SFT from an ON-POLICY RL perspective in gradient space and absorb all additional terms into the reward, we directly avoid the demonstration-distribution problem and, from the reward angle, reveal SFT’s potential issues and improve SFT, and validates them with extensive experiments across a range of tasks.
>
> We find the blog’s derivation from an alternative perspective highly valuable. We have cited and discussed it directly in the revised version of our paper to reflect its relevance and contribution to this line of research.
>
> **Q2. Compared with RL method**
>
> Table 1 is designed to focus specifically on the comparison between vanilla SFT and DFT, in order to isolate and highlight the improvements brought by DFT itself. The goal is to evaluate DFT’s effectiveness without confounding it with additional RL-specific components.
>
> We do provide direct comparisons between DFT and several RL-based methods, including DPO, RFT, PPO, and GRPO in Table 2 of the paper. The results show that DFT can achieve performance comparable to these advanced RL algorithms under the SFT setting. Furthermore, when combined with rejection sampling to obtain denser reward signals, DFT is able to match or even surpass these baselines on several mathematical reasoning benchmarks.
>
> **Q3. Can DFT be combined with RL?**
>
> DFT can be used as an initialization method for RL. We evaluate three tasks using models initialized with either SFT or DFT before applying GRPO training. The results demonstrate that DFT offers a stronger initialization and consistently yields better performance after GRPO training. We have updated the results in the paper (Table 5-7). Thank you for your suggestion.
>
> | Model | Math500 | Minerva Math | Olympiad Bench | AIME24 | AMC23 | Avg. |
> |-------|---------|---------------|----------------|--------|--------|-------|
> | Qwen2.5-Math-1.5B w/ SFT+GRPO | 62.54 | 23.10 | 26.92 | 5.00 | 40.15 | 31.54 |
> | **Qwen2.5-Math-1.5B w/ DFT+GRPO** | **65.96** | **23.51** | **28.37** | **8.63** | **41.40** | **33.57** |
>
> | Model | HumanEval | HumanEval+ | MultiPL-E |
> |--------|-----------|-------------|-------|
> | Qwen2.5-Coder-3B w/ SFT+GRPO | 57.3 | 50.6 | 58.15 |
> | **Qwen2.5-Coder-3B w/ DFT+GRPO** | **68.9** | **61.0** | **62.61** |
>
>
> | Model | MathVerse | MathVision | WeMath |
> |--------|----------------------|-------------|---------|
> | Qwen2.5-VL-3B w/ SFT+GRPO | 35.93 | 21.44 | 21.43 |
> | **Qwen2.5-VL-3B w/ DFT+GRPO** | **39.06** | **23.35** | **26.19** |
>
>
>
> Besides, thanks for sharing the paper "Simplify RLHF as Reward-Weighted SFT: A Variational Method", it’s indeed highly relevant to our topic, and we’ve actually discussed it before.
>
>
>
> We hope our clarifications help resolve the concerns you raised.

---

> > ### Comment · Reviewer_JU9o · 2025-11-27
> >
> > I thank the authors for the responses and updates. I am happy to see the discussion about the ICLR blog in the paper. I am leaning towards an accept so my rating of 6 is justified at the moment. I would like to do a final pass on authors-reviewers discussion at the conclusion of discussion period and update my rating if that's justified.

---

> > > ### Author Response · Authors · 2025-11-27
> > >
> > > Thank you for your valuable feedback and for your positive recognition of our updates. Your comments have been very helpful in improving our work and the manuscript.

---

### Author Response · Authors · 2025-11-29
**Summary of Discussion Points for the Area Chair**

Dear AC,

We sincerely appreciate your time and effort in handling our submission. We also thank all three reviewers for their thoughtful assessments and for recognizing the simplicity, practicality, and empirical effectiveness of Dynamic Fine-Tuning (DFT).  Here is the summary of discussion points during the rebuttal period:

- Across the initial reviews, key concerns focused on clarifying the intent behind the RL-based reinterpretation of SFT (`JU9o` and `d8uu`), improving conceptual precision in the theoretical motivation (`d8uu`), examining the boundary conditions under which DFT may underperform (`8xgv`), and providing further analysis on calibration, stability, and alternative weighting strategies (`8xgv`). Reviewers also asked for expanded comparisons with RL baselines and an evaluation of whether DFT can serve as a stronger initialization for downstream RL training (`JU9o`).

- In our rebuttal, (1) we refined the positioning of DFT as a method inspired by RL perspectives but operationally compatible with SFT, and clarified that the RL formulation is intended purely as an analytical tool rather than a literal description of SFT’s mechanics. We revised the theoretical narrative to clearly distinguish intuition from formal statements and removed language that could be interpreted as overstating sparsity or instability. (2) A dedicated Limitations section has been added to clarify scenarios where DFT is less effective, particularly the acquisition of factual knowledge, supported by new experiments on the Natural Questions dataset. (3) We also included extended comparisons against RL methods such as DPO, RFT, PPO and GRPO, and showed that DFT consistently provides stronger initialization for RL across mathematical reasoning, coding and multimodal tasks. (4) Additional analyses on calibration effects, sequence-level weighting variants and stability considerations were also incorporated.

- After reading our rebuttal, Reviewer `JU9o` expressed satisfaction with the clarifications and leaned toward acceptance. Reviewer `d8uu` maintained a positive assessment after the expanded limitations analysis. Reviewer `8xgv` confirmed that the concerns were resolved. We appreciate the reviewers’ recognition of the strengthened motivation, expanded empirical evaluation and refined theoretical framing.

We have incorporated all clarifications, new analyses and updated results into the revision. With these improvements, our work is strengthened and made more solid in both motivation and empirical support.

Thank you again for taking the time to read our summary.

Best regards,

Authors of Submission 5472

---

### Meta-Review · Area_Chair_gfhG · 2025-12-28

**Summary:**

Reviewers agreed the proposed token-probability reweighting of the SFT cross-entropy shows strong empirical performance and is simple to implement, but raised concerns about: (i) missing comparisons evaluating whether DFT helps downstream RL fine-tuning, (ii) insufficient discussion of limitations and failure modes, and (iii) the possibility that the RL-based theoretical framing is imprecise/overstated and could mislead readers about “instability/sparsity” claims that arise primarily from a change-of-measure rewrite (d8uu). The first two concerns were addressed during rebuttal and acknowledged by the reviewers. The last concern raised by Reviewer d8uu remains largely outstanding (see below).

I have carefully read the reviews, discussion, and paper, and I largely agree with Reviewer d8uu’s assessment: while the empirical results are strong, the theoretical framing/motivation is currently the weakest part of the submission. In particular, the RL-based narrative (SFT -> RL -> SFT) feels artificial and risks obscuring a more direct interpretation: the method is naturally viewed as a reweighted cross-entropy objective, closely related in spirit to objectives such as focal loss (with an inverted weighting philosophy). The authors acknowledge this connection, but it is currently relegated to the related-work section; in my view, it could serve as a more central and straightforward motivation, with the RL perspective presented as a heuristic lens  for motivating the specific choice of weighting by p rather than as a principled derivation.

Relatedly, the paper’s positioning versus weighted-CE objectives and some broad narrative claims (e.g., the “LLM era” shift toward overfitting/memorization dominating objective design) remain under-justified and should be toned down or scoped carefully. The added limitations section is an important and constructive addition, and I encourage the authors to build on it in the final version.

More generally, the “theory” claims should be checked and tightened. For example, the step from Eq. (8) to (9)) is better described as a stability-motivated heuristic: it is not an exact algebraic transformation, and the connection to PPO/importance sampling is not clear as stated (it reads more like a switch from sequence-level to token-level reweighting). I recommend that the final version explicitly clarifies this approximation and adjusts the associated justification/citations accordingly.

Overall, acknowledging the positive scores, the strong empirical performance, the partially successfull rebuttal, I recommend acceptance with a strong encouragement to the authors to take the recommendations of Rev. d8uu (and above) into account.
I hope the authors find this feedback helpful in strengthening an already practically useful contribution.

**Reviewer Concerns:**

explicitly stated above

**Reviewer Scores:**

I believe all three reviewers would have maintained their weak acceptance scores 6.

---

### Decision · Program_Chairs · 2026-01-26

Accept (Poster)